# The effect of age on visuomotor learning processes

Chad Michael Vachon[1,2], Shanaathanan Modchalingam[1,3], Bernard Marius 't Hart[1]*,
Denise Y. P. Henriques[1,2,3]

**1** Centre for Vision Research, York University, Toronto, Ontario, Canada, **2** Department of Psychology, York University, Toronto, Ontario, Canada, **3** School of Kinesiology and Health Science, York University, Toronto, Ontario, Canada

* thartbm@gmail.com

**Data Availability Statement:** All data files are available from the Open Science Framework (DOI: 10.17605/OSF.IO/QZHMY).

**Funding:** This work was supported by a Canadian Network for Research and Innovation in Machining

## Abstract

Knowing where our limbs are in space is essential for moving and for adapting movements to various changes in our environments and bodies. The ability to adapt movements declines with age, and age-related cognitive decline can explain a decreased ability to adopt and deploy explicit, cognitive strategies in motor learning. Age-related sensory decline could also lead to a reduced fidelity of sensory position signals and error signals, each of which can affect implicit motor adaptation. Here we investigate two estimates of limb position; one based on proprioception, the other on predicted sensory consequences of movements. Each is considered a measure of an implicit adaptation process and may be affected by both age and cognitive strategies. Both older (n = 38) and younger (n = 42) adults adapted to a 30˚ visuomotor rotation in a centre-out reaching task. We make an explicit, cognitive strategy available to half of participants in each age group with a detailed instruction. After training, we first quantify the explicit learning elicited by instruction. Instructed older adults initially use the provided strategy slightly less than younger adults but show a similar ability to evoke it after training. This indicates that cognitive explanations for age-related decline in motor learning are limited. In contrast, training induced much larger shifts of state estimates of hand location in older adults compared to younger adults. This is not modulated by strategy instructions, and appears driven by recalibrated proprioception, which is almost twice as large in older adults, while predictions might not be updated in older adults. This means that in healthy aging, some implicit processes may be compensating for other changes to maintain motor capabilities, while others also show age-related decline (data: https://osf.io/qzhmy).

## Introduction

In order to move successfully, we need to know where our limbs are in space. In order to adapt our movements to changing circumstances, we should also update how we estimate where our limbs are. It has been shown that the ability to adapt movements declines in healthy aging, but the effect of age on limb position estimates is still unclear. Both age-related sensory decline

Technology NSERC Operating grant (DYPH) and
the German Research Foundation (DFG) under
grant no. HA 6861/2-1 (BMtH). The funders had no
role in study design, data collection and analysis,
decision to publish, or preparation of the
manuscript.

**Competing interests:** The authors have declared
that no competing interests exist.

and age-related cognitive decline may provide partial and interacting explanations. We evoke motor adaptation with a visuomotor rotation and primarily test a sensory explanation by gauging proprioception-based shifts in hand location estimates, in the context of prediction-based shifts of hand location estimates. To test if age-related cognitive decline factors into hand-location estimates, we also manipulate strategy-based adaptation. The goal of this study is to understand how aging contributes to changes in all these processes.

Older adults can adapt their reaches to altered visual feedback of the hand, but in some cases have slower or incomplete learning compared to younger adults. We observe that age-related effects are mostly found in studies using larger cursor rotations (such as 60˚ or 90˚), where older adults do not adapt at the same rate nor to the same extent as younger adults [1–7]. A similar difference between age-groups can be seen in force-field adaptation [8, 9] as well as prism adaptation [10] although see [11] for an alternative finding. However, when the visual perturbation is small (such as 30˚), age-related differences in initial stages of motor learning are less robust: with some studies showing a deficit for older adults [12] and others not [4, 7, 11, 13], sometimes within the same study [14]. Since larger rotations tend to evoke more explicit, cognitive adaptation than smaller rotations do [15–17], age-related cognitive decline may explain differences between age groups that are reported with larger rotations [4, 18]. Here we intend to test if age-related changes in post-adaptation changes in limb-position estimates are due to age-related sensory changes, which affect the fidelity of error signals, but we also test if the availability of cognitive, explicit strategies has any influence on limb-position estimates and if this is different in younger and older participants.

Adaptation is traditionally thought to solely rely on implicit processes, which Stanley & Krakauer [19] suggest is a misunderstanding of seminal work, for example on patient H.M. [20]. While the idea is older [21], in the last decade or so, explicit processes have been established as a substantial contributor to motor learning, including visuomotor adaptation [4, 22–25]. The explicit component of motor learning is often assessed by having participants indicate their aiming strategy either on a trial-by-trial basis during adaptation, often verbally [4, 26], or indicate their strategy post-adaptation by reaching without a perturbation or cursor [4, 5, 15, 16, 27, 28]. In addition, explicit instructions about how to compensate for a visual perturbation are used to facilitate initial learning [15, 26, 28, 29]. The cognitive nature of the explicit contribution is further demonstrated by decreased performance when adapting to a force-field while doing a secondary, distracting task [30–32] although no clear effect has been shown for visuomotor adaptation [33, 34]. While small perturbations may be compensated for entirely implicitly, perhaps up to ~15˚ of rotation [35], larger perturbations additionally recruit explicit mechanisms [17], resulting in cognitively accessible strategies [15, 16]. This also explains why the explicit processes emerge early in training when the reaching errors from the perturbation are largest and easiest to detect [26]. The resulting reach aftereffects that follow adaptation to all sizes of perturbations reflect these implicit motor and likely sensory changes [16, 36]. In sum, while explicit and implicit processes both contribute to motor adaptation, explicit processes crucially rely on cognitive abilities which may change with healthy aging.

Thus, changes in cognitive processes due to healthy aging may affect explicit motor processes [14]. This hypothesis is somewhat supported by the work of [37–39]. Using a verbal aiming-strategy report before and after training with a 75˚ rotated cursor, they found that older adults used their strategy less, as they underestimated the angles needed to reach to the target with the perturbation. However, this group did not find this effect with a 30˚ rotated cursor. Some studies suggest that for younger adults, lower spatial working memory or performance on a secondary memory task is correlated with poorer learning rates [1, 30, 40]. From this literature [2], it could be inferred [7] that age-related cognitive decline [41, 42] can also explain age-related differences in adaptation rates. While Noohi et al. [13] find a lower

awareness of the perturbation in older adults, this might be an issue with the questionnaire's construct validity, as the same participants show no effect of age on adaptation to a 30˚ rotation. In a more recent, comprehensive study, Vandevoorde & Orban de Xivry [14] show decreased explicit learning in older adults and find that this was compensated with implicit learning–which may also explain the findings of Noohi et al. [13]. However, other groups have shown that implicit learning is limited [17, 35], so that implicit adaptation may only compensate for deficits in explicit learning to a limited extent or older adults may have an increased capacity for implicit learning. In this study, we will use visuomotor adaptation and first assess the extent that aging affects the ability to initially adopt and later maintain explicit strategies and then gauge different implicit contributions.

If a decrease in explicit learning in older adults is compensated by increased implicit learning, we should not only be able to measure this in reach aftereffects, but also in shifted estimates of hand location. Visuomotor adaptation leads to shifts in estimates of hand location, that can be based on proprioception and on predicted sensory consequences, and we consider both to be implicit. Proprioceptive estimates of hand location have been robustly shown to shift, or be recalibrated, following both force-field adaptation [43, 44] and visuomotor adaptation [45–52] where hand location estimates shift in the direction of the visual feedback on hand location by ~20% of the visuomotor distortion [45–52]. In turn, this visually-induced change in hand proprioception appears to contribute to changes in motor performance, particularly reach aftereffects [53–57]. These changes in proprioception seem to be largely implicit, in that they are not affected by instructions [16], or by introducing the rotation abrupt or gradually [58] Proprioceptive recalibration also does not change with age when using the usual perceptual reports [45, 56]. Thus, if implicit processes compensate for age-related declines in cognitive or sensory processes, this may affect proprioceptive recalibration. While previous findings predict that proprioceptive recalibration should not differ between older and younger adults, we will test this here with a different measure.

Like proprioception, efferent-based hand position estimates are an important contributor to state estimation and are central to adaptation. In fact, many computational models of motor learning specify that predicted sensory consequences based on efferent signals, must be updated for implicit motor learning to occur [59]. The amount of change in these state estimates has been quantified by measuring changes in hand localization following visuomotor adaptation [16, 47, 49, 60, 61]. Synofzik et al. [60] and Izawa et al. [61] find that cerebellar patients show reduced but substantial adaptation-induced changes in hand location estimates, about 50% of those in healthy controls and conclude that the reduction reflects the critical role that the cerebellum plays in updating predicted sensory consequences for learning. Further, the remaining shifts in hand localization in these patients suggest that the part of these learning-induced changes that may be due to proprioceptive recalibration could occur outside the cerebellum [56]. The contributions of the afferent and efferent sources of these changes in hand localization have been recently investigated in young adults [16, 47, 49, 62]. However, it is not clear how predicted estimates of hand location change with healthy aging, or if they are more affected by (compensate for) either age-related cognitive- or sensory changes, so that we will test this here.

Here we investigate the extent to which age-related deficits in outcomes of motor learning–both reach aftereffects and hand localization shifts–could be explained by the distinct effects of age on explicit adaptation and the implicit contributions of shifts in proprioceptive and predictive estimates of hand locations to implicit adaptation. To do so, we use a 30˚ rotation which allows us to evoke explicit learning in half the participants by providing them with instructions [16, 29]. This way we can directly assess the differences between adaptation with and without explicit contributions, specifically how this affects shifts in hand location estimates. In

particular, we first test if there are any age-related differences in cognitive strategies during and after motor adaptation by assessing the impact of instruction in both older and younger adults. We also assess whether there are age-related differences in implicit contributions to no-cursor reaches. Finally, and most importantly, we test whether aging affects shifts in proprioceptive and predictive hand location estimates as components of implicit adaptation. Our results provide a more comprehensive understanding of how age affects the contributions of explicit and implicit processes to visuomotor adaptation, primarily the adaptation-induced changes in hand location estimates.

## Methods

### Participants

Forty-one younger (mean age of 20.9 years, SD = 2.77) and thirty-eight older (mean age of 70.0 years, SD = 6.78) adults were recruited through various research participant pools at York University: the Undergraduate Research Participant Pool (URPP), Kinesiology Undergraduate Research Participant Pool (KURE), York Research Participant Pool (YRPP), and also from the surrounding community. The participants from the undergraduate research pools were given course credit for participation. Older adults who were recruited through the YRPP or the community were paid a honorarium and lunch, as required by the YRPP, to compensate for time and travel to the university. All participants had normal or corrected-to-normal visual acuity, were right handed, and self-reported that they were in good health and were able to understand the tasks. Participants gave their informed written consent before taking part in the study. The York Human Participants Review Sub-committee approved this study (#2014–240).

All participants adapted to a 30˚ visuomotor rotation, however just before the rotated session, half of each age group were given explicit instructions on the nature of the perturbation, and a strategy to counter the rotation, while the other half received no instructions. Thus, there were four groups: non-instructed younger adults (n = 20, 14 female), instructed younger adults (n = 21, 13 female), non-instructed older adults (n = 19, 6 female) and instructed older adults (n = 19, 14 female). The instructions were meant to result in a cognitively accessible strategy, although participants could also spontaneously develop a cognitive strategy. The difference between the instructed and non-instructed groups in the availability of a strategy was tested by asking people to reach with or without their strategy–whether or not this strategy was obtained from instructions or spontaneously developed. Then finally, we could compare hand localization shifts between older and younger adults, gauging the influence of explicit adaptation on hand localization shifts in the two age groups.

We excluded 4 younger (all from the non-instructed group) and 3 older (1 non-instructed, 2 instructed) participants. One older participant didn't complete the experiment. All other excluded participants often "leaned on" the robot manipulandum during the 300 ms pre-reach hold period (see below) so that when the handle was released they made a sudden hand movement toward themselves and away from the targets. That is why, in these participants, reach direction relative to the target direction could not be determined for a lot of trials. The data of the excluded participants was never fully pre-processed and hence not statistically analyzed nor included in the online data set.

### Setup

Participants were seated in front of a table on a height-adjustable chair so that they could comfortably see and reach to displayed targets projected on a reflective surface from a monitor (Samsung 510 N, 60 Hz), located 28 cm above a 2-joint robot manipulandum (Interactive

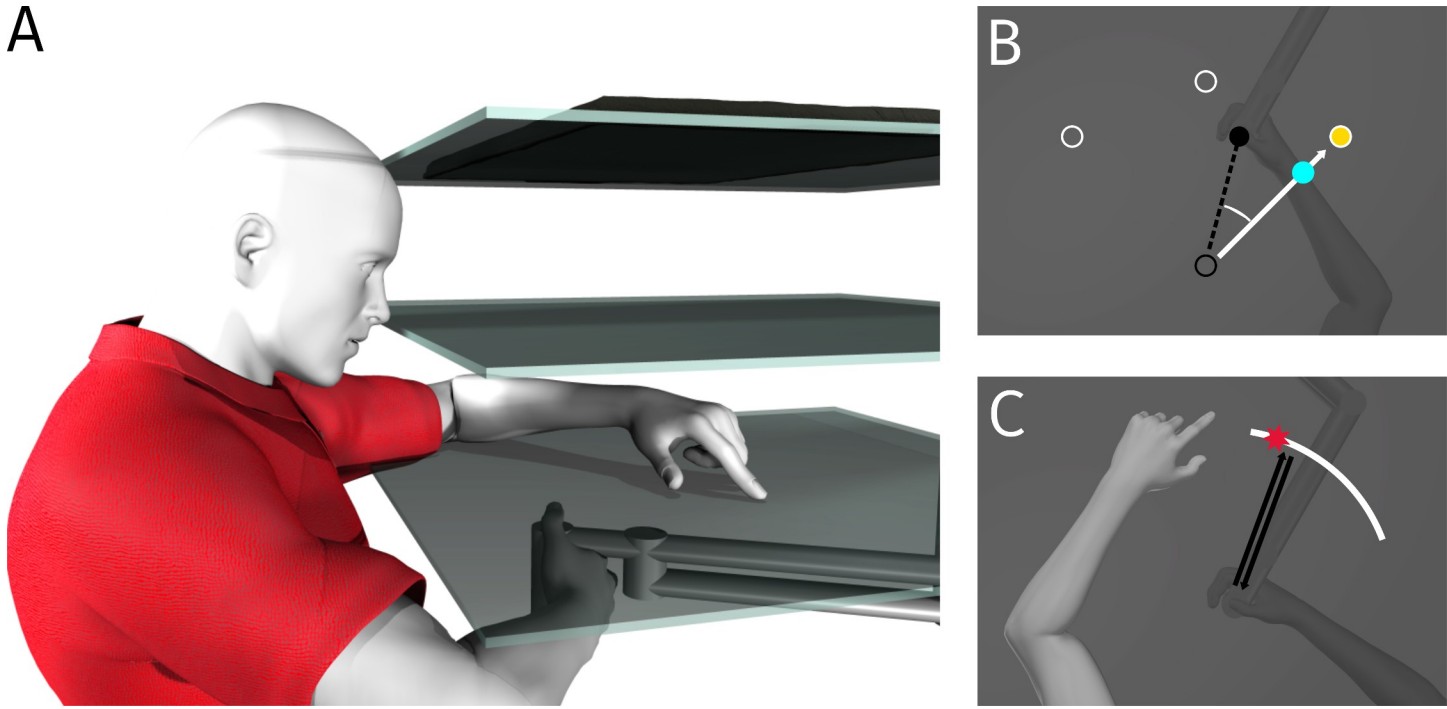

**Fig 1. Apparatus and experimental setup.** A: Participants moved their right hand while gripping the handle of a two-joint robot manipulandum. Stimuli are produced by a downward-facing monitor (top surface) above a reflective surface (middle). The touchscreen panel is located below the reflective surface and just above the robot handle (bottom). B: Training task: The three white hollow circles represent the targets. They can either be directly in front of the home position, 45° clockwise (CW) or counter clockwise (CCW) from it. During the main rotated reach training, the cursor was blue and rotated 30° CW (white solid vector) relative to the actual (unseen) hand position/direction (black dashed vector). C: Localization task: Using 't Hart and Henriques [33] localization task, the unseen right hand was moved to a location along a white arc either by participants voluntarily moving their hand or having their hand physically moved by the robot. Once participants returned their right hand to the home position, they indicated on the touchscreen with their visible left hand, where their trained right hand had intersected the arc (illustrated by red star).

Motion Technologies Inc., Cambridge, MA, USA). The reflective surface was 14 cm above the robot manipulandum and 14 cm below the downward facing monitor (Fig 1A). The chair was fixed in its position for the duration of the experiment. Participants were asked to grip a vertical handle with their right hand. They were instructed to place their right thumb on a screw located on top of the robot handle. The handle was attached to the free end of the robot manipulandum and could be moved on a horizontal plane. A thick black cloth was draped and tucked over participants' right shoulder to ensure that they did not see their right arm.

Since the reflective surface was located halfway between the monitor and the robot manipulandum, the reflection of the cursor and the targets appeared on the same plane as the thumb of the right hand. The reflective surface also occluded the right hand and arm.

During some tasks, participants used their visible left hand, which was then illuminated by a lamp, to respond on a touchscreen panel (Keytec Inc., Garland, TX, USA sampled at a resolution of 1440 × 900 pixels) located 2 cm above the robot handle, to indicate the perceived position of their unseen right thumb (Fig 1A and 1C).

## Procedure

The procedure and set up are similar to our previous study [16], and could be completed in 90 minutes (excluding breaks and instructions). It includes two experimental sessions (Fig 2), and each session had several repetitions of the same four tasks (described in detail below). The first session (illustrated on the top row of Fig 2) measured baseline results where the cursor

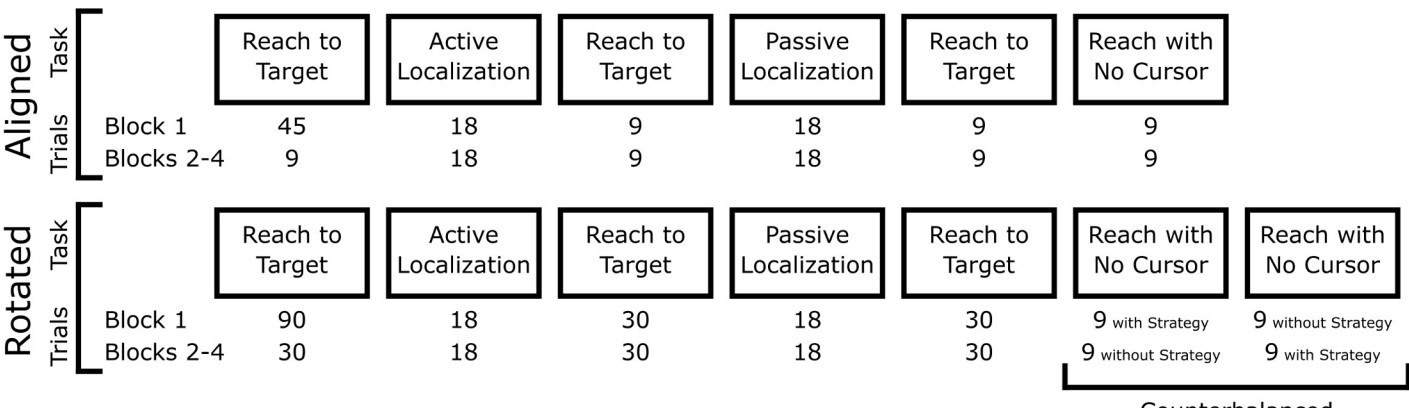

**Fig 2. Task order.** Top row: First session was the baseline session, where the cursor was aligned with the hand position in reach training tasks. Bottom row: Second session involved training with a rotated cursor. Each session began with reach training and consisted of four tasks (e.g. reach training, active localization, passive localization, and no cursor reaches). Each set of four tasks represents a block, and each block was completed four times, starting with training. This was followed by two hand localization tasks, during which either the participants moved their own unseen hand to the arc (active localization) or the robot guided their hand (passive localization). "Top-up" reach training followed each of these localization tasks. Each block ended with a no-cursor reach task. For the no-cursor reaches following training with a visuomotor rotation in the second session (bottom), participants completed this task twice, once when told to not use their strategy (no-cursor reaches without strategy) and once when told to use it (no-cursor reaches with strategy). The order of these two tasks were counterbalanced within and between participants.

was aligned with the hand during reach training. The second session (bottom row) had training with a 30° CW rotated cursor as illustrated in Fig 1B. Initial reach training was followed by several other tasks (see Fig 2) to measure training-induced changes, interleaved with additional training to counter decay. Each block of tasks was repeated four times in each session. During the break between the two sessions, half of the participants were instructed about the nature of this 30° perturbation in the second session while the other half were not. Some older participants took additional breaks, as desired, but only right before a reach training task. After completing both sessions of the experiment, participants were then asked a series of questions to assess awareness of the perturbation (https://osf.io/qzhmy/).

Before each task started, a short reminder of what to do was displayed. Each task consisted of a number of trials, and involved one of several kinds of hand displacements. At the beginning of each trial the hand was locked into place for 300 ms at the home position (~20 cm in front of the participant on their body midline). While the hand was locked in place, either a target or an arc was displayed. After the hand displacement trial, participants returned their hand to the home position along a constrained straight path, generated by a perpendicular resistant force of 2 N/(mm/s) and a viscous damping of 5 N/(mm/s), to begin the next trial. The different tasks are described below.

**Training.** During reach training tasks (Fig 1B), participants' hand position was represented as a cursor (1.0 cm in diameter) which was green when it moved in alignment with the hand, and blue when it represented the hand position rotated 30° around the home position. At the beginning of the trial the hand was locked into place, while a target was displayed as a yellow dot (1.0 cm in diameter) located 12 cm straight in front of the home position (at 90°) or diagonally to the right (45°) or to the left (135°). After 300 ms the hand was released, and the participant was to move the cursor to the target, i.e. get the center of the hand cursor within 0.5 cm of the target's center. Then the target and cursor disappeared and participants moved their hand back to the home position along a constrained straight path, to begin the next trial.

In the first session, (top row of Fig 2), the cursor was aligned with the hand position. The first aligned training task included 45 trials. In between the other tasks (localization and no-cursor tasks), additional blocks of 9 identical training trials were performed. In the second

session (bottom row of Fig 2), the cursor represented the hand position, but rotated by 30˚ CW relative to the home position. The first rotated training task included 90 trials while subsequent rotated training tasks consisted of 30 trials to keep adaptation saturated.

**Instructions.** For both the older and younger adult groups, half of the participants were instructed on a strategy to counteract the 30˚ CW cursor rotation, and the other half did not get this instruction. Instruction was provided using an animation as well as a verbal explanation and a clock diagram (used in [29]), where 1 hour represents 30˚ (see https://osf.io/qzhmy/ ). It was ensured that participants understood these instructions; i.e., that they could draw, for at least three times, an arrow on a clock illustrating the direction at which they would reach when using the provided strategy. The non-instructed participants were told that "the reach training tasks will be different as the cursor will not move the same way as it did previously, and they will need to compensate for it by figuring out an appropriate strategy". Both instructed and non-instructed participants were also told they would be called upon to use their strategy during some subsequent reaches and not use the strategy in other reaches (described in detail below).

**No-cursor reaches.** Participants reached to the same three targets as in the training trials (see above) but without visual feedback about the position of their hand (i.e. no cursor). Trials were considered complete when participants had moved away from the home position and indicated that they had acquired the target by holding their hand still for 500 ms. Then the target disappeared, and participants moved the robot handle back to the home position, through a constrained channel, to begin the next trial. During the first, aligned session, the no-cursor reach tasks involved asking participants to reach nine times in each of four tasks (for a total of 36 trials; 12 to each target). During the second, rotated session, these no-cursor trials were split into two sub-tasks (of nine trials each). In one sub-task, participants viewed the words "No-Cursor with Strategy" displayed on the monitor and were required to use the strategy that they also used during their recent training with the blue cursor. In the other subtask, participants saw the words "No-Cursor without Strategy" and they were asked to not use a compensatory strategy and reach as they did in the aligned session (as in [15]). The order of these subtasks was counterbalanced between and within each consecutive participant: some started with strategy and some without and this order was inverted in subsequent no-cursor tasks for every participant. Each pair of subtasks was repeated four times (for a total of 72 trials; 24 to each target, 12 of which with and 12 without strategy).

The results of the no-cursor reaches with and without strategy were compared as per a process dissociation procedure (PDP; see [15]), to determine the levels of explicit and implicit learning. The no-cursor reaches without strategy reflect implicit learning (involuntary reach aftereffects) while any additional deviation in the direction of no-cursor reaches with strategy reflects explicit learning. This procedure was used to determine if participants were aware of the nature of the rotation, and could apply the strategy at will, even if they had no visual feedback of their hand. This allowed testing if the instructions lead to explicit learning, and if this was different for the two age groups.

**Localization tasks.** The localization tasks assessed the participants' estimated location of their unseen hand following reach training, when they either reached with their hand themselves (active localization) or when the robot displaced their hand (passive localization) to the white arc (Figs 1C and 2). Both active and passive localization allowed the use of afferent information on hand position. Since participants generated a motor command in active localization, they *also* had efferent information on hand location in this localization task [16, 47, 49]. By measuring both, we can tease apart the contribution of (and changes in) afferent and efferent estimates for hand localization.

In the active localization tasks, participants made a quick and straight hand movement with the robot manipulandum toward a chosen point on a white arc located 12 cm away from the home position. The arc was 0.5 cm wide, spanned 60˚ and was put in three locations: centred on 50˚, 90˚, or 130˚ in polar coordinates (Fig 1C shows the 50˚ position). When the robot handle reached the distance of the arc, a force "cushion" was applied to prevent the participant from moving their hand past the arc, giving them the sensation of hitting a soft wall. After "hitting the wall", participants returned their right hand to the home position (along the same constrained path as in the other tasks), and then used their visible, and untrained left hand to indicate on the touch screen the point where they thought the right hand intersected the arc (Fig 1C). To avoid unwanted contact with the touch screen, participants placed their left hand under their chin between each response. Six hand localization trials were completed at each of the arc locations (for a total of 18 trials) for each of the four repetitions (Fig 2) which sum to a total of 72 trials per session.

In the passive localization tasks, the participants' unseen right hand was pulled by the robot manipulandum to a specific point on the arc on each trial. These points were the endpoints of the reaches generated by the participant in the preceding active localization task. Like in the active localization task, participants returned their hand to the home position along a constrained path and then indicated with their visible left hand, the location of the point on the arc where the right hand had been (Fig 1). This was done for the same number of trials as in the active localization task.

## Data processing

For all trial types, angular deviations were calculated. For training trials, this was the angle between a line through the home position and the target, and a line through the home position and the position of the hand at maximum velocity on the outward part of the reach. For no cursor trials, the position of the hand at the end of the reach was used instead. For localization trials, the angle between a line through the home position and the actual position of the unseen right hand and a line through the home position and the location indicated with the seen left hand on the touch screen was calculated.

The hand path and velocity profile of every single reach trial (with cursor or without) and the endpoint for every localization trial were visually inspected for quality to remove trials with obvious measurement or task errors, such as failure to reach the target, understand task instructions, or localization responses that did not land near the visually presented arc as required. We removed slightly fewer trials from younger adults' data (training: 1.4%, no-cursor: 1.2%, localization: 1.1%) than from older adults' data (training: 2.0%, no-cursor: 2.6%, localization: 1.9%), but percentages are low overall (more details: R notebook).

All statistical analyses were done using R 3.4.4 (R Core Team, 2018). All statistical tests used an alpha level of 0.05, and for ANOVAs where sphericity was violated, the Greenhouse-Geisser corrections were used. Partial eta-squares (denoted $\eta^2$) were used to report effect size for significant effects in ANOVAs. Significant interactions were followed up with a Welch Two Sample t-test (correcting for unequal variances), with eta-squared as effect size (also denoted $\eta^2$).

## Results

### Rotated reach training

Before investigating the localization shifts, we first wanted to confirm that the effect of instruction (instructed or non-instructed) on initial reach adaptation was similar for older and younger adults (Fig 3). To do this we compared the reach deviations during the first three reach training trials, second set of three trials and the final set of 15 trials of the 90 trials in the initial

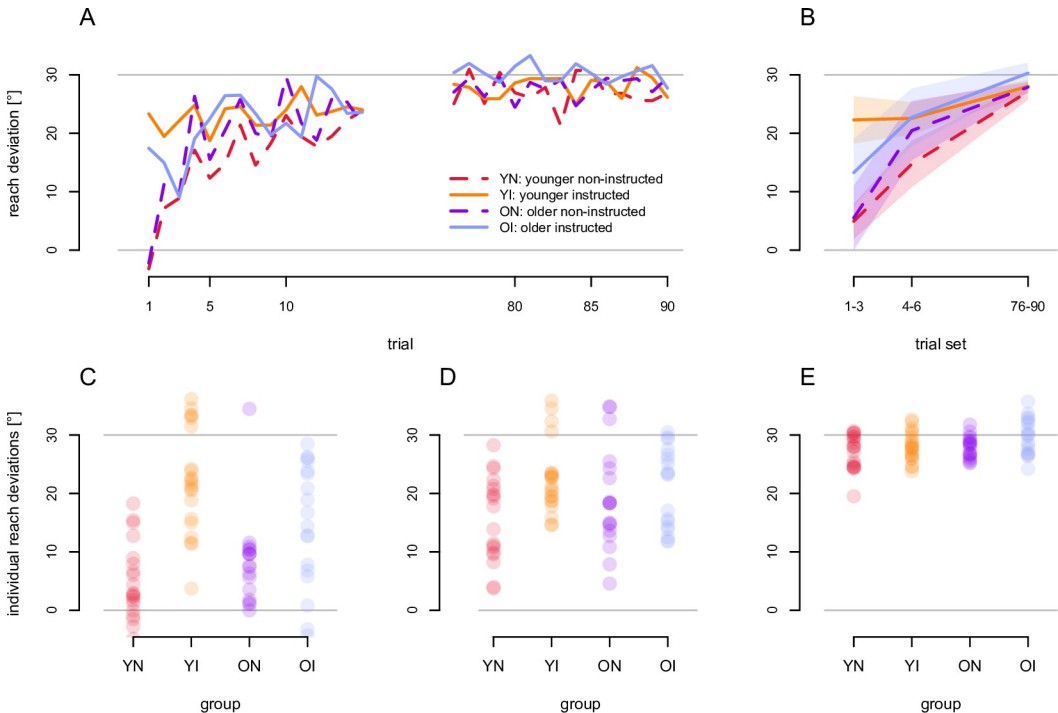

**Fig 3. Learning curves.** Lines represent the mean, baseline-corrected angular reach deviations for each group with instructed groups as solid lines, and non-instructed groups as dashed lines. To fully counter the rotation, angular reach deviations should be 30°. Shaded areas represent 95% confidence intervals. Dots represent individual participants' reach deviations. A: Mean reach deviations are shown across the first 90 training trials with rotated cursor. B: The mean across participants and trials within the initial, second and final trial sets. C, D, E: Individual participants' reach deviations in the first (C), second (D) and final (E) trial set.

rotated-cursor training task (with the reach deviations from the last 15 trials in the initial aligned training task subtracted). We performed a 2 x 2 x 3 mixed ANOVA with age (younger and older) and instruction (instructed or non-instructed) as between-subject factors and trial set (first, second and final) as a within-subject factor. As shown in Fig 3, all participants' reach deviations, regardless of instruction and age, significantly increased across trial sets (Fig 3B), as required to compensate for the cursor rotation. This was verified by a main effect of trial set, $F(2, 150) = 94.81$, $p < 0.001$, $\eta^2 = 0.48$, There was an interaction between trial set and instruction, $F(2, 152) = 10.45$, $p < 0.001$, $\eta^2 = 0.082$, showing that instruction affected the time course of adaption. More importantly, there was an interaction between age and instruction, $F(1, 75) = 4.63$, $p = 0.035$, $\eta^2 = 0.022$, showing that the ability to use instructions or not across trial sets to compensate for the perturbation does depend on whether the participant was older or younger.

As prior research showed that instruction, or explicit strategies, mainly affected adaptation during early training [26, 29], we compared age (older and younger) and instruction (instructed and non-instructed) for the initial two sets of trials separately in two follow-up ANOVAs. For the first trial set, a 2 x 2 ANOVA with instruction (instructed and non-instructed) and age (older and younger) on reach deviations. We found a main effect of instruction, $F(1, 75) = 32.32$, $p < 0.001$, $\eta^2 = 0.301$, as well as an interaction between age and instruction, $F(1, 76) = 4.72$, $p = 0.033$, $\eta^2 = 0.059$. In a planned follow-up t-test on reach deviation for the first trial set the older and younger instructed participants showed an effect of age, $t(33.34) = 2.71$, $p = 0.011$, $\eta^2 = 0.16$. That is, we found that reach deviations in instructed older adults (violet line in Fig 3A & 3B) were one-third smaller than those in instructed younger adults (orange line). A second planned t-test on

instructed and non-instructed older adults (Fig 3A & 3B, solid blue and dashed purple lines, respectively) showed an effect of instruction within older participants, $t$ (35.96) = 2.06, $p$ = 0.047, $\eta^2$ = 0.106. In summary, in the first three training trials, reach deviations were larger (greater cursor compensation) for the instructed groups compared to the non-instructed ones, but these were not as large for the instructed older adults compared to the instructed younger adults.

We also tested the age-dependent difference in reach deviations as a function of instruction for the second set of trials using a 2 x 2 ANOVA using age (older and younger) and instruction (instructed or non-instructed) as between-subject factors. Instruction still led to overall greater reach deviations, $F$ (1, 75) = 6.29, $p = 0.014$, $\eta^2$ = 0.077, which suggests instructions still provided a benefit in training trials 4, 5 and 6. However, this benefit of instructions did not vary with age, $F$ (1, 75) = 1.89, $p$ = 0.17, $\eta^2$ = 0.024. And as illustrated in Fig 3, by the last trial set, all groups attained near perfect compensation (reach deviations of roughly 30˚).

## No-cursor reaches

No cursor reaches were used to assess cognitive awareness of the cursor rotation developed during training with a rotated cursor both as a function of age, instruction (both factors as above) as well as strategy use (with strategy or without strategy). First, we determined if adapting to a visuomotor rotation led to changes in no-cursor reach directions, or reach aftereffects, by comparing no-cursor reach directions without strategy from the rotated session with no-cursor reach deviations in the aligned session. As we expected, significant reach aftereffects of approximately 15˚ did emerge, $F$ (1, 75) = 746.93, $p < 0.001$, $\eta^2$ = 0.75. Given that training with a visuomotor rotation did produce significant changes to no-cursor reaches, i.e. there were reach aftereffects, we continued analyzing no-cursor reach deviations with the aligned no-cursor reach deviations subtracted from both the with-strategy and without-strategy no-cursor reach deviations.

To confirm similar explicit adaptation in older and younger adults, we ran a 2 x 2 x 2 ANOVA on reach aftereffects, with age and instruction as between-subjects factors (as used above) and strategy use as within-subjects factor (see Fig 4). For adaptation to be consider to have an explicit component, it should at least be possible for people to either use that part of adaptation (their strategy) at will or not. In other words, for those aware of the cursor rotation,

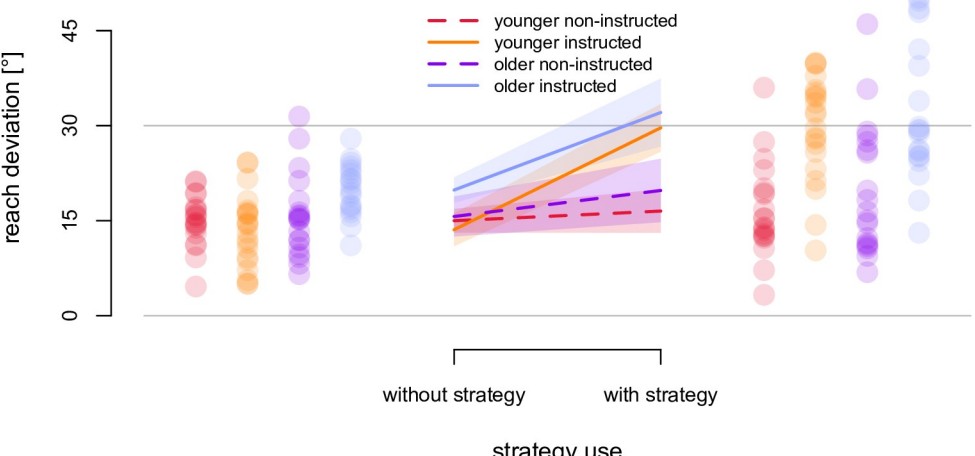

**Fig 4. Changes in no-cursor reach direction following training.** Shown are mean, baseline-corrected angular reach deviations in no-cursor reaches for each group, i.e, reach aftereffects, while suppressing (left side) or employing (right side) any strategies used during adaptation. Full reach aftereffects would be a ~30˚. Shaded areas and error bars are 95% confidence intervals. Dots represent individual participants' reach aftereffects.

we expect the corresponding no-cursor reach deviations when asked to reach with a strategy to be larger than those when asked not to use the strategy. And, for those who are not aware, there should be no difference between these two no-cursor reach tasks. We used this process dissociation procedure [15, 16] (PDP), to determine whether this measure of awareness varied with age. There was a main effect of age $F(1, 75) = 5.66$, $p = 0.02$, $\eta^2 = 0.07$, which could suggests older adults had larger with- and without-strategy reach aftereffects, and relied more on implicit learning. This was not systematically the case however; as Fig 4, and post-hoc tests showed (see R notebook), this effect was small and driven solely by the instructed older adults reach deviations in the without-strategy condition. We also found an interaction between instruction and strategy use, $F(1, 75) = 35.56$, $p < 0.001$, $\eta^2 = 0.12$. This suggests that only those given instructions produced larger no-cursor reach deviations when asked to use the strategy (right side of Fig 4) when compared to not using the strategy (left side of Fig 4). More importantly, we found no difference in this pattern between older adults (blue and purple) and younger adults (orange and red), as there was no three-way interaction between age, instruction and strategy use on no-cursor reach deviations, $F(1, 75) = 2.86$, $p = 0.09$, $\eta^2 = 0.012$. This suggested that the effect of instruction on the availability of an explicit strategy was comparable for both age groups.

In other words, despite smaller reach deviations for instructed older adults during the first trial set in rotated training, instructed older adults did evoke the strategy during the no-cursor reaches when asked. Crucially, it seemed that older instructed adults could evoke their explicit strategy to the same extent as younger instructed adults. This meant that instructions evoked explicit learning and they did so equally for younger and older adults.

## Hand localization

We then tested if afferent and efferent estimates of hand location varied as a function of explicit learning and if this was different for older and younger participants (Fig 5). First, we confirmed that training with a rotated cursor induced shifts in hand localization using a four-way 2 x 2 x 2 x 2 mixed ANOVA that included training session (aligned and rotated session) and movement type (active and passive localization) as within-subject factors, and age (older and younger) and instruction (instructed and non-instructed) as between-subjects factors. There was a main effect of training, $F(1, 75) = 231.34$, $p < .001$, $\eta^2 = 0.28$. This meant that the estimated location of the unseen hand in the rotated session was significantly shifted relative to the aligned session, and in the direction expected after training with a CW rotated cursor. Moreover, training session significantly interacted with movement type, $F(1, 75) = 10.01$, $p = .002$, $\eta^2 = 0.002$. Thus, the results replicated prior findings [47] of a significant effect of movement type; that is, the training-induced shifts in hand localization were larger for active localization than for passive localization, confirming an additional contribution to state estimates of hand location from updated efferent-based predictions of sensory consequences in active localization.

Given that the rotated cursor training led to a significant change in hand position estimates, which in turn varied with movement type, the next step was to test whether age and instruction affected this pattern. First, like for the reach aftereffects, we took the difference in localization between the rotated session and the aligned session, to calculate hand localization *shifts* for both the active and passive localization tasks (as plotted in Fig 5A, 5B, 5D and 5E). We tested whether these shifts in hand-localization varied as a function of age and instruction, using a 2 x 2 ANOVA for passive localization, where only afferent (proprioceptive) information was available (Fig 5B). Older adults (purple and blue lines) produced significantly larger shifts in the afferent-based estimates of hand position, almost double that of younger adults (orange

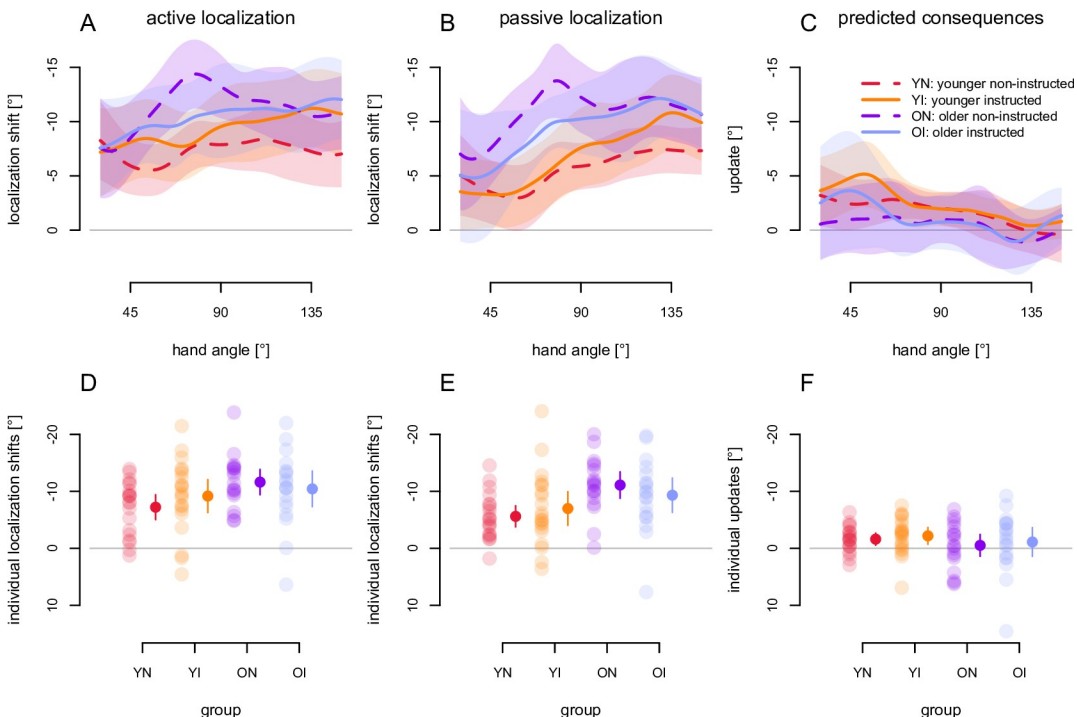

**Fig 5. Training-induced shifts in hand localization.** A: Active localization: mean, interpolated angular shift in hand localization estimates after training where participants actively moved their unseen right hand to a location of their choosing on the white arc (efferent and afferent). B: Passive Localization: Mean shift in angular hand localization estimates after training where the unseen right hand is passively moved to the exact location as the active localization task (only afferent). C: Predicted sensory consequences: Differences in estimates between active hand localization (efferent and afferent) and passive hand localization (afferent only) are interpreted as evidence there are changes in efferent-based estimates of hand location. Shaded areas are 95% confidence intervals. D, E, F: Individual participants' average of the kernel smoothed localization shift at 50°, 90° and 130° (used for statistics) as transparent dots, with the group average and 95% confidence interval as opaque dots and lines, for active localization (D), passive localization (E) and the difference between active and passive localization (F) as estimate for an added contribution of predicted sensory consequences to hand localization.

and red lines), $F(1, 75) = 10.01$, $p = 0.002$, $\eta^2 = 0.118$, but there was no main effect of instruction, $F(1, 75) = 0.02$, $p = 0.87$, $\eta^2 < 0.001$, suggesting that explicit learning did not affect afferent hand localization signals. Lastly, there was no interaction between instruction and age, $F(1, 75) = 1.63$, $p = 0.21$, $\eta^2 = 0.021$. Thus, in summary, while older adults showed larger shifts in afferent-only hand localization, or proprioceptive recalibration, there was no effect of instruction on proprioceptive recalibration, confirming this is a wholly implicit process [16].

As indicated earlier, the shifts in hand localization were larger in active localization (Fig 5A) than in passive localization (Fig 5B), likely because this task also includes changes due to updated efferent-based or predicted estimates of hand position. However, when we tried to isolate the updated predicted or efferent-based estimates by subtracting shifts in hand localization responses in the passive condition (Fig 5B) from those in the active condition (Fig 5A), the difference (Fig 5C) and the effect size were small overall. This suggests that the training-induced changes in hand estimates in the active localization task (Fig 5A) might be primarily due to proprioceptive recalibration (Fig 5B).

Further, as in the original 4-factor ANOVA a 2 x 2 ANOVA on the difference between active and passive localization shifts (as in Fig 5C) using age and instruction as between-subject factors, we found no effect of age, $F(1, 75) = 1.59$, $p = 0.19$, $\eta^2 = 0.02$ (nor any other effect or interaction). However, when we compared the magnitude of these updated predicted shifts

in hand position for each age group to zero, older adults showed only 0.84˚ shift, which did not significantly differ from 0, $t(35) = 1.07$, $p = 0.15$, $\eta^2 = 0.032$. However, the shifts for younger adults, 1.97˚, were significantly different from 0, $t(41) = 4.62$, $p < .001$, $\eta^2 = 0.481$, as found by 't Hart & Henriques [47]. These results were consistent with the confidence intervals shown in Fig 5C. In summary, while older adults showed much greater shifts in afferent-based hand localization, or proprioceptive recalibration, than younger adults, they did not show evidence of a contribution from updates to efferent-based predictions in hand localization.

## Discussion

In this study, we assess the effects of age on several processes in motor learning. Specifically, we gauge the effects of age on implicit shifts in proprioceptive and predictive hand localization as well as how these are modulated by explicit, strategy-based adaptation. We find that older adults benefit less from explicit instruction during initial adaptation although they demonstrate a similar amount of strategy as younger adults when reaching without a cursor after training. In contrast, no-cursor reaches without a strategy show no impairments in implicit changes for older adults that could contribute to age-related decline in motor learning. In fact, instructed older adults show marginally larger without-strategy reach aftereffects. Older adults also show clearly larger proprioceptive recalibration, which contributes to increased hand localization shifts compared to younger adults, suggesting that specific implicit processes are more heavily relied on with advanced age. In addition, while there is an added effect of efferent-based signals on hand localization shifts in younger adults, this is not detectable in older adults. In summary, while we find some suggestion of age-related deficits in the initial use of an explicit strategy, we neither find deficit in the ability to evoke the strategy after training, nor a clear effect of age on implicit motor changes. Nevertheless, we see age-related implicit changes in hand localization, as we find that proprioceptive recalibration is larger in older adults.

### Explicit component of adaptation in aging

Our findings replicate prior research that demonstrates that instruction on how to counteract a perturbation can benefit initial adaptation [15, 16, 26–29]. Here we find that the extent of this benefit decreases as a function of age by about ⅓ (Fig 3). Similarly, Heuer & Hegele [4, 27] demonstrate that, when given both instruction and corrective feedback, older adults are less able to acquire or apply a cognitive strategy when training to rotate an arrow to match the direction they would have to reach to counter a visuomotor rotation. However, since they average over performance throughout learning the time course of the effect is unclear. Extending this, we show that age-related differences in applying a cognitive strategy are limited to only the initial stages of learning (specifically the first set of 3 trials), albeit for a smaller rotation of 30˚. Our findings seem to indicate that the ability or willingness to adopt a novel explicit strategy decreases with age, while the ability to apply a learned strategy is not affected. How this affects older participants' ability to adapt to larger rotations remains to be seen.

Other age-related deficits might emerge even in the absence of instruction, when perturbations are large enough [2–7, 10, 63], which has implications for cognitive awareness and processing of the visual perturbation [15, 28]. For smaller rotations (e.g. 30˚ rotation) the findings are not consistent: in the absence of instructions some studies find age-related deficits in adaptation [12, 63] while others find no effect [4, 7]. This may be understood if implicit adaptation is limited in scope [17, 35, 64]. The increased implicit adaptation in older adults in a task designed to only elicit implicit adaptation [14] shows it is also possible that older adults' increased capacity for implicit adaptation in turn reduces the overall need and capacity for

explicit strategies. Either way, explicit, cognitive strategies are necessary to counter larger, but not smaller, rotations. For example, using a 75˚ rotation, Heuer and Hegele [27, 37, 38] find that fewer older adults show a difference between including and excluding a strategy when asked, and when they do, the difference is smaller. Taken together, these studies suggest a decreased ability in older adults to spontaneously develop cognitive strategies for larger rotations [14].

Several studies propose that the age-related deficits in motor learning may be due to cognitive decline as a function of age. Earlier work suggests at least three possible cognitive mechanisms: spatial working memory [1, 65], which could be used when applying a strategy to aim for a different location [22]; inhibition [66], which may be necessary to suppress regular, unadapted reach planning; and divergent/convergent thinking [67], which may play a role in discovering and fine-tuning a cognitive strategy during explicit learning. Our data here can not provide or test any partitioning of cognitive mechanisms that do or do not contribute to explicit motor learning. But, if fine-tuning of a strategy depends on convergent thinking [67], an age-related deficit in convergent thinking is consistent with the smaller benefit of instruction we see in older adults' early adaptation. In their review of this literature, Buszard & Masters [18], suggest that indeed the evidence relating cognitive performance and visuomotor adaptation performance is weak, but stronger for sequence learning [13]. Sequence learning is used to assess skill acquisition rather than what we test here; skill maintenance. Since maintenance of existing skills is arguably fundamental for everyday life, it could make sense if this faculty is prioritized over cognitive mechanisms necessary for larger perturbations and acquisition of new skills in advanced age. Despite valuable exploratory work [1, 63, 66, 67], the identification of the cognitive processes that are both affected by age and involved in motor adaptation and motor learning remains tentative and incomplete.

## Implicit components of adaptation in aging

The aim of this study is not to investigate the mechanisms underlying explicit, cognitive adaptation, but to test the effects of age on shifts in hand localization and if these are modulated by the availability of an explicit strategy. We first look at reach aftereffects and find little to no effect of age on without strategy no-cursor reach deviations after training (implicit reach aftereffects). Since some studies have already found that implicit reach aftereffects are not affected by explicit learning [16, 26], it is not surprising that we find little effect of instruction on implicit reach aftereffects, except perhaps a slight increase in the without-strategy reach aftereffects in instructed older adults. We also find that both younger and older adults can evoke an equal explicit strategy at will in no-cursor reaches. This means that any effects of age on hand localization shifts are not due to differences in implicit adaptation and that different effects of instruction on hand localization shifts in the two age groups are not due to different availability of explicit strategy. Below, we will consider two contributions to shifts in hand localization: recalibrated proprioception and updated predictions of sensory consequences.

**Proprioceptive recalibration and aging.** In contrast to implicit reach aftereffects, we do see a clear effect of age on localization throughout, which seems to be driven by recalibrated proprioception, with no evidence of an increase of predicted sensory consequences in older adults. While an earlier study from our lab shows proprioceptive recalibration in older and younger adults to be the same, roughly 6˚ [45], here we find it is larger in older adults (10.2˚) as compared to younger adults (6.3˚). That previous study uses a two-alternative, forced-choice paradigm where participants indicate whether their hand was clockwise or counter-clockwise from a visual reference with two interleaved staircase procedures across fifty trials to find the point of subjective equivalence. It could be that the faster method of measuring hand

localization here is less susceptible to decay (cf.: [51, 68]) and that this especially affects measurements in older adults. Apart from the methodological differences, Cressman et al., [45] also have fewer older participants (9 vs. our 38) which are slightly younger (average 66 years vs. 70 years). Consequently, increasing the sample size by roughly four times, combined with using a slightly older population in this study perhaps provides the necessary power to detect a difference between age groups. Nonetheless, we find that the implicit change in proprioceptive hand estimates following learning is at least not smaller than in younger adults, and therefore cannot explain possible age-related decline in adaptation, although it may be part of compensation for age-related changes in other processes.

Another explanation for increased proprioceptive recalibration in older adults could be that older adults rely more on visual over proprioceptive feedback. However, this does not seem to be a good explanation for our results. Findings from Block & Bastian [69, 70] show no relationship between sensory weighing and sensory realignment following training with visual-proprioceptive mismatches in healthy, younger adults [70] and in cerebellar patients as well as their age-matched (i.e. older) controls [69]. In addition, other studies measuring estimates of final hand positions also fail to find a correlation between hand estimates using multiple modalities and the precision or relative reliability of either visual or proprioceptive-only estimates of location [71–73]. While there are studies that suggest that proprioceptive acuity is poorer in older adults than younger adults (see [74] for a review), including one that tests felt hand position in our own lab [45], the current study, with our much larger sample of participants and larger number of proprioceptive trials, does not find that older adults are less precise in their estimates of hand position prior to training with a rotated cursor [75]. Moreover, when compared across all these older and younger adults, the size of proprioceptive recalibration does not correlate at all with the amount of variance in hand-estimates [75]. Together, these studies demonstrate that a possibly greater reliance on vision resulting in greater proprioceptive recalibration in older adults cannot simply be explained by poorer proprioceptive sensitivity [76]. Further studies are required to understand the factors that influence the amount of proprioceptive recalibration.

**Updated prediction with age.** Motor learning does not only lead to changes in implicit processes such as proprioceptive estimates of hand position, but also to updates in predicted sensory consequences–a required change to produce implicit adaptation. 't Hart and Henriques [47] found that proprioception accounted for at least half of learning-induced changes in hand localization (as originally found by [60, 61] using only an active localization task), compared to prediction. This partitioning out of predictive and proprioceptive based change is consistent with the results of the cerebellar patients in the studies by Izawa et al. [61] and Synofzik et al. [60]. Both studies find that the patients show a significant change in active hand localization following adaptation, but that the change in hand estimates is smaller than in healthy controls. Both studies interpret their result to support the idea that the cerebellum is critical for updating the predicted consequences of movements during learning. This leaves open the possibility that the remaining localization shift found in the cerebellar patients may be due to proprioceptive recalibration, which our lab has shown to also occur in cerebellar patients [56]. In the current study, the relative proportion of predicted and perceived changes in hand localization was even more skewed than those of 't Hart and Henriques [47] (despite more training targets and more localization trials), with younger adults showing a change in hand localization: 80% of which was due to a change in proprioception and 20% a change in prediction. The change in predictive estimates of hand position is further reduced in older adults, such that the ~1˚ change is not statistically detectable (although their proprioceptive change is 50% larger than that of younger adults). This difference is not significant relative to younger adults since the changes for younger adults are also quite small; approximately

20% of the total localization shift reflecting only a 2 to 3˚ shift (Fig 5C and 5F). It may be that the proprioceptive recalibration of larger magnitudes in older adults may act as a ceiling effect or generally mask any further change due to prediction. Nonetheless, given that we find no clear age-related effect on updates of predicted estimates–which are implicit by nature and not affected by instruction (as shown in [16])–we can not definitively claim here that decreased updates in predicted estimates explain age-related effects on motor adaptation, but it does remain a possibility. If updates of predicted sensory consequences do decrease in old age, this could hint at age-related deficits in the cerebellar contribution to motor adaptation. This would also explain the absence of a difference between active and passive localization we see in our findings. Conversely, in order to get hand location estimates to match visual feedback as much as in younger adults, this might spur an increase in proprioceptive recalibration, as we observe in this study.

The clearest effect we find in this study is the age-related increase in hand-localization shifts in older adults that is not modulated by the availability of a strategy. Whether or not this underlies a larger reliance on implicit learning or is an effect of reduced proprioceptive acuity in older adults remains to be seen. Other perspectives assume a deficit in explicit adaptation that in turn necessitates a larger reliance on implicit adaptation. However, some studies find that implicit adaptation is limited to about what we find here [35], so that implicit learning would not be able to compensate for loss of explicit adaptation in advanced age. However, with the same paradigm that should only result in implicit learning, i.e. no modulation by explicit learning should be possible, older adults still adapt more [14]. This hints at another possibility: with advanced age, we have an increased capacity for implicit adaptation [77], in turn reducing the need for explicit adaptation or we simply forego explicit adaptation as we learn from experience that this is often wasted effort. All this work also suggests that the additive nature of the push-pull dynamics between explicit and implicit processes that many studies tacitly assume, may just be overly simplistic. For example, in the current work, as in some of our other work [16, 36], we can clearly elicit explicit adaptation, but this does not modulate implicit adaptation. Here we see that in advanced age, the composition of implicit processes contributing to adaptation may change. Of course, with larger perturbations, explicit processes are engaged more [16, 17], and this is where age-related deficits in motor adaptation are clearer. To overcome this, e.g. for rehabilitation purposes, the focus of motor learning research should perhaps be on how to improve or increase implicit contributions to motor learning and motor adaptation, especially in an aging population.

## Conclusion

In this study we keep explicit contributions to motor adaptation comparable in both older and younger participants, in order to test age effects on hand localization shifts. We find larger hand localization shifts in older adults compared to younger adults, and these are largely due to an increase in proprioceptive recalibration. In contrast, updated predictions of sensory consequences may or may not be lower in older adults but can not explain age differences. Our data are in line with the hypothesis that decreased proprioceptive acuity leads to an increase in implicit adaptation in advanced age. However, given the slightly slower adoption of the strategy provided by our instructions, it may also be that with advanced age, people rely more on automatic implicit processes rather than effortful explicit learning. Regardless, proprioceptive recalibration is increased in older adults, and this shows that aging may lead to subtle, yet important changes, in sensorimotor learning and multisensory integration.

## Author Contributions

**Conceptualization:** Bernard Marius 't Hart, Denise Y. P. Henriques.

**Data curation:** Chad Michael Vachon, Shanaathanan Modchalingam.

**Formal analysis:** Chad Michael Vachon, Bernard Marius 't Hart.

**Funding acquisition:** Denise Y. P. Henriques.

**Investigation:** Chad Michael Vachon, Shanaathanan Modchalingam, Bernard Marius 't Hart.

**Methodology:** Bernard Marius 't Hart, Denise Y. P. Henriques.

**Project administration:** Chad Michael Vachon, Shanaathanan Modchalingam, Bernard Marius 't Hart, Denise Y. P. Henriques.

**Resources:** Denise Y. P. Henriques.

**Software:** Bernard Marius 't Hart.

**Supervision:** Bernard Marius 't Hart, Denise Y. P. Henriques.

**Validation:** Chad Michael Vachon, Shanaathanan Modchalingam, Bernard Marius 't Hart, Denise Y. P. Henriques.

**Visualization:** Chad Michael Vachon, Bernard Marius 't Hart.

**Writing – original draft:** Chad Michael Vachon.

**Writing – review & editing:** Chad Michael Vachon, Shanaathanan Modchalingam, Bernard Marius 't Hart, Denise Y. P. Henriques.

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
