## [Decision Letter · Decision Letter 0]

6 Mar 2020

PONE-D-19-19971

The effect of age on visuomotor learning processes

PLOS ONE

Dear Dr. 't Hart,

Thank you for submitting your manuscript to PLOS ONE. After careful consideration, we feel that it has merit but does not fully meet PLOS ONE’s publication criteria as it currently stands. Therefore, we invite you to submit a revised version of the manuscript that addresses the points raised during the review process.

Please, review the comprehensive reviews and make an effort to place this study within the body of previous work on the same subject. It is important to reconcile the differences in the outcomes. The questions related to the statistical issues in the study design should be addressed. Also, please, follow the suggested improvements of the description.

We would appreciate receiving your revised manuscript by Apr 20 2020 11:59PM. To enhance the reproducibility of your results, we recommend that if applicable you deposit your laboratory protocols in protocols.io, where a protocol can be assigned its own identifier (DOI) such that it can be cited independently in the future. For instructions see: http://journals.plos.org/plosone/s/submission-guidelines#loc-laboratory-protocols

We look forward to receiving your revised manuscript.

Kind regards,

Sergiy Yakovenko

Academic Editor

PLOS ONE

Journal Requirements:

Reviewers' comments:

Reviewer's Responses to Questions

**Comments to the Author**

1. Is the manuscript technically sound, and do the data support the conclusions?

Reviewer #1: Yes

Reviewer #2: Yes

Reviewer #3: Yes

2. Has the statistical analysis been performed appropriately and rigorously? 

Reviewer #1: Yes

Reviewer #2: Yes

Reviewer #3: Yes

3. Have the authors made all data underlying the findings in their manuscript fully available?

Reviewer #1: Yes

Reviewer #2: Yes

Reviewer #3: Yes

4. Is the manuscript presented in an intelligible fashion and written in standard English?

Reviewer #1: Yes

Reviewer #2: Yes

Reviewer #3: Yes

5. Review Comments to the Author

Reviewer #1: SUMMARY

The manuscript describes a study aimed to evaluate the effects of instruction and strategy use on visuomotor rotation ability during reaching movements and, importantly, how instruction and strategy use might interact with age. Their sample included older adults (mean age 20.9 years) and younger adults (mean age 70.0 years). All participants were adapted to a force field using an arm manipulandum. Some participants received instruction from the experimenter on how to counteract the force field and some participants did not. To assess implicit learning, the experimenters assessed aftereffects after adaptation. To assess sensory and non-sensory contributions, they measured each participant’s ability to locate the place where their hand had been while moving the manipulandum after adaptation. The manipulandum was either moved by themselves (active) or by a robot (passive) and they were not able to see the hand being moved. Finally, to assess explicit learning, the experimenters used the between-participants instruction manipulation (instructed, not instructed) that occurred before adaptation and added a within-subjects strategy use manipulation (use strategy, don’t use strategy). They asked participants to either use or not use a strategy while interacting with the manipulandum after adaptation. All measures were compared to a baseline that was the use of the arm manipulandum with no force field applied. During the first trials of adaptation blocks, instruction was more effective in younger adults than older adults during adaptation – younger adults who received instruction adapted faster than older adults who received instruction. After adaptation, aftereffects were similar between age groups, suggesting that implicit learning was similar in older and younger adults and unaffected by instruction. Active arm reaches with the manipulandum led to larger proprioceptive realignment (based on hand localization) than passive arm reaches, and this active/passive effect was greater in the older adults than in the younger adults. Instruction did not affect hand localization. Finally, they found an interaction between instruction and strategy after adaptation such that participants who received instruction from the experimenter in how to counteract the force field performed better than participants who received no instructions when they used that strategy compared to when they did not. Age did not interact with instruction and strategy use, suggesting that both age groups apply explicit instruction similarly after adaptation.

Taken together, their results suggest that older adults may have more difficulty implementing an instruction about how to adapt their arm movements when initially adapting to a new visuomotor mapping. This difficulty may be specific to the early stages of adaptation because they found that older adults apply strategy similar to younger adults after adaptation stabilizes and as the effects of adaptation begin to fade.

This is a generally well-written and methodologically sound manuscript. Most of my comments are minor. I have four overall comment followed by more specific comments.

OVERALL COMMENTS

#1 As I understand it, the premise set forth by the authors is this: Older adults are slower at adapting to large rotations than younger adults. There is no difference between older and younger adults at adapting to small cursor rotations (although there is some evidence to the contrary). Together, the two findings suggest the hypothesis that older adults have preserved implicit processing with deficits in explicit processing. In support of this hypothesis, they note that there is no difference between older and younger adults in reach after-effects after adaptation, a process widely considered to be implicit.

To draw a parallel between large/small rotations and explicit/implicit processing, the authors need to provide more information: To what extent are larger cursor rotations truly driven by explicit processes compared to implicit processes? To what extent are smaller cursor rotations truly driven by implicit processes compared to explicit processes?

The authors address this missing information with statements, such as these:

Lines 57 – 48: “Adaptation requires both a cognitive, or explicit, component which tends to contribute to early stages of learning, as well as implicit processes that predominate in the later stages.”

Lines 67 – 68: “Because larger rotations produce larger initial reaching errors than smaller rotations, they are more likely to require and evoke cognitive processes…”.

They do not provide citations, however, or explanations of empirical work that would support the large/small rotations and explicit/implicit parallel.

This is very important, because the purpose of the study was to understand how age is related to explicit and implicit processes during reach adaptation.

#2 My second overall comment is related to #1. In the Introduction and Discussion, the authors indicated that age group differences between older and younger adults in adaptation are consistently found with large rotations (between 60 and 90 degrees), but not with smaller rotations. They hypothesize that this age group difference is due to a greater reliance upon explicit processes in older adults than in younger adults.

The stated purpose was to better understand how age is related to explicit and implicit processes during adaptation.

Why, then, did the authors chose a small rotation (30 degrees) for adaptation?

Additionally, much of the literature review in the Introduction as well as the literature referred to in the Discussion is about studies with large rotations. Would it be possible to bring in studies that used small rotations, given that this study used a small rotation?

#3 The Methods, Results, and the first paragraph of the Discussion were extremely clear. There are many places in the Introduction and Discussion, however, that were a bit more difficult to read. I think that some of the difficulty can be attributed to the presence of irrelevant details that made it hard to immediately see the most relevant point. Here is an example:

Lines 510 – 513: “Both of which have been shown to be unaffected by instruction or cognitive strategy [16,18] and are similar in magnitude whether the distortion is introduced gradually or abruptly (as demonstrated across different studies using the same setup; e.g., [46,57]).”

The portion in italics is less relevant (maybe not at all relevant?) and, therefore, adds confusion to the sentence and paragraph. Please consider a revision of the Introduction and Discussion that removes these add-ons. This will make the reading of those sections much easier because the reader will not have to filter out these less relevant parts to see the take-home point of the sentence/paragraph.

#4 The Abstract does not appear to capture the same take home of the study that is presented in the first paragraph of the Discussion. The Abstract seems to focus on the active/passive results while the majority of the manuscript seems to focus on the explicit instruction/strategy use results. Please spend some time making the take homes presented in the Abstract and Manuscript more cohesive with one another.

ABSTRACT

Line 3: Although grammatically correct, it sounds strange to follow “adaptation of movements” with “is”. Might be better to say, “Movement adaptation … is”, or, “Adaptation of movement … is”.

Lines 28 – 30: The sentence starting with, “Following visuomotor adaptation” feels rather opaque. Could it be simplified/clarified?

INTRODUCTION

Line 35: “persistent changes” is a confusing word pair. It’s clear what you mean after a few rereads, but it would be better if confusing word pairs like that were not in the first sentence. Maybe something more like, “Our brain has evolved to adapt our movements to the environment and our body. Our movements are able to adapt to changes online as well as changes that persist in time.”

Lines 36 – 37: “Reach adaptation is based on both explicit and implicit processes.” Please provide a citation for this statement. Please also add some text that provides some indication of what you mean by the words explicit and implicit.

Line 44: “although not for [11].” Should include the example to which they are referring, e.g., “although not for prism adaptation [11]” or “although see [11] for an alternative finding”.

Line 47 should start a new paragraph.

Line 57 should start a new paragraph.

Lines 57 – 58: I hesitate a bit here: “Adaptation requires both a cognitive, or explicit, component which tends to contribute to early stages of learning, as well as implicit processes that predominate in the later stages.” It is possible that it only feels more explicit during the initial portion of adaptation – there are several studies that suggest that visuomotor actions, especially ballistic actions like reaching, can be performed entirely implicitly, where implicitly means unconsciously, and are rarely, if ever, affected by explicit processes, where explicit means conscious (e.g., work of Milner and Goodale). It’s possible that my hesitation here could be addressed by providing your definition of the terms “implicit” and “explicit”.

Relatedly, does the literature referenced here have any bearing on whether adaption is affected by a participant’s explicitly stated strategy or whether the participant’s explicitly stated strategy is, in fact, related to the experience of the adaptation effect?

Lines 67 – 69: “Because larger rotations produce … to evoke cognitive processes…” The logic of this statement is not clear. Please clarify.

Lines 71 – 72: Missing citation for the sentence beginning with, “This is in contrast …”.

Lines 74 – 75: “… explicit and implicit processes appear to contribute to different aspects of adaptation performances.” The literature review that precedes this statement does not support this statement well. Please clarify the preceding literature review so that this conclusion statement is well supported.

METHODS

Line 175: missing period at end of sentence

Figure 2 is a bit confusing because it seems to suggest that the presence/absence of instruction on the perturbation was manipulated within-participants (… because it’s easy for the reader to make the mistake of equating ‘strategy’ and ‘instruction’). Is there a way to make the figure clearer? Maybe add asterisks that state that the “strategy” could have been given to them (i.e., instructed) or made up on their own (i.e., not instructed). Maybe adding some clarification on this point throughout the paper would help, too. I initially made the mistake of equating ‘strategy’ with ‘instruction’, and it took me a while to figure that out!

How easy/difficult/possible is it for a participant to not use a strategy once they have practiced it? In other words, how plausible are the ‘strategy’ and ‘no strategy’ conditions in practice?

Why do the authors not consider the strategy that the participants made up on their own a sort of explicit instruction? They assume the participant’s own strategy is solidified enough that the participant can either use it or not use it because this is part of the test design. It seems to me that the difference is “explicit strategy developed by another person” and “explicit strategy developed by oneself”.

Line 267: typo, “leaning” should be “learning”

RESULTS

The results plots are lovely. It was a bit difficult for me to keep in my mind which measure they were depicting, despite the name of the measure being in the image. I would recommend adding a more explicit statement in the figure caption about how the measure was calculated or, if feasible, adding a schematic that depicts that calculation of the measure to the figure. Just a suggestion.

The ordering of the subsection is not consistent with the ordering of the subsections in the Methods sections. “No Cursor Reaches” is described last in the Methods, but it is not last in the Results. The order in the Methods is best, because it is more coherent with the timeline of the experimental procedures. Please reorder the subsections so that they are in the same order in the Methods and Results sections.

Lines 327 – 331: “For those aware of the cursor rotation, the corresponding no-cursor reach deviations when asked to reach with a strategy should be larger than those when asked not to use the strategy. And, for those who are not aware, there should be no difference between these two no-cursor reach tasks. We used this process dissociation procedure (PDP), to determine whether this measure of awareness varied with age.”

This is very concise, but it needs to be expanded a bit. It’s not clear to me how the effectiveness of their strategy is an indication of their awareness.

This is particularly important given the interaction between instruction and strategy that is reported and the conclusion starting on line 340, “… the effect of instruction on awareness was equal for both age groups.” It is possible that participants were equally aware of the perturbation in all conditions and the interaction was because the experimenter-provided strategy (i.e., instructed condition) was simply more effective than the strategies that the participants came up with on their own (i.e., not instructed condition). The conclusion would, then, be, “… the effect of instruction on the effectiveness of the strategy was equal for both age groups.”

The Results section moves between present and past tense often. Past tense is standard. I would recommend a quick read through to ensure that it is in past tense. Here are a few that I caught, but there are likely others:

Line 357: “interacts” should be “interacted” and “replicates” should be “replicated”

Line 359: “are” should be “were”

Line 362: “leads” should be “led”

Line 363: “varies” should be “varied”

DISCUSSION

This initial paragraph is quite strong (starting line 397). It very clearly explains and synthesizes the results. At the beginning of this paragraph the authors say, “we find clear evidence for age-related deficits in “explicit” aspects of adaptation”, but by the end they say, “we find some suggestion of age-related deficits in the use of an explicit strategy.” These two statements appear to contradict one another a bit. Please clarify. I would suggest removing the word “clear”.

Lines 407 – 409: The sentence starting with, “However”, uses a double negative at the end, “… as to be non-detectable in older adults but not younger adults” and is, consequently, difficult to read/comprehend. Please reword.

Lines 414 – 416: The sentence starting with, “Using …” has something wrong with it grammatically. Or, maybe it’s just missing a comma? Please edit.

Line 421: “demonstrate” should be “demonstrated”

Line 428 – 429: “This suggests that the ability or willingness to adopt a novel explicit strategy decreases with age.”

Is it possible that the ability or willingness to adopt a novel explicit strategy—at any age—could have some dependence on the magnitude of the rotation? It seems that your results really only speak to explicit/implicit processes at small rotations because you did not test at large rotations. You vaguely mention this in line 430 with, “… under certain perturbations.” Could you make it more explicit that the certain perturbation to which you are referring is 30 degrees?

Lines 465 – 505: I don’t think this section is needed. Consider removing or refocusing it on the results of the current study. As it is, it seems more like background information for a study correlating cognitive measures with adaptation. I can see how it is related, but it is not needed and doesn’t add much to the discussion, in my opinion.

Reviewer #2: This manuscript tested the effects of instruction and strategy use on how well older and younger adults were able to compensate for a 30-degree visuomotor rotation during reach-training and then use this strategy afterwards when reaching without a cursor. Training-induced changes in proprioceptive and predicted estimates of the adapted hand in the two age groups were then compared. They found that instruction benefitted older adults less than younger adults during initial training, but that older adults exhibited a similar pattern in reach aftereffects, suggesting that older adults’ strategy use could be evoked during no-cursor reaches after enough training. They also found that implicit changes (proprioceptive recalibration) and reach aftereffects in older adults were greater than those in younger adults, independent of their awareness of the rotation, perhaps due to age declines in proprioceptive acuity. From these results, the authors suggest that the explicit contributions to motor learning decrease with age, whereas the implicit processes remain intact.

In my view, this is an interesting and well-written manuscript with clear and motivated hypotheses. I do, however, have a list of questions / general concerns:

• Of the 4 younger and 3 older adults who were removed due to task incompletion, how many were in the non-instructed vs. instructed groups?

• Regarding the experimental procedure, how long was each session?

• Page 21, line 296, “We find…” should be “We found…”

• Pages 26-27, starting on line 424, “Using the same instructions…”. This sentence doesn’t make sense; is it supposed to be two sentences? Also, should it be “we find that instructed younger participants compensate more”?

• Was the age difference in recalibration with age significantly larger in older vs young adults?

• Although the authors offer an explanation based on previous studies that cannot explain the age difference in proprioceptive calibration, they do not really go into detail about potential explanations that can. Can the authors speculate as to why implicit processes might be enhanced with aging? Could this be due to compensatory mechanisms with aging? What kinds of experimental procedures could future studies use to understand the factors that influence the amount of recalibration?

• In the first sentence of the conclusion, “This study demonstrates that age-related decline leads…” Decline in what? Cognitive function?

Reviewer #3: The authors have investigated age differences in implicit and explicit processes of sensorimotor adaptation. They found little effects of age on adaptation, but older adults exhibited greater effects on felt hand position post practice.

Vandervoorde & Orban de Xivry (2019 Neurobio Aging) have already investigated age differences in implicit and explicit adaptation processes. Noohi et al. (2016 Neuropsychologia) also address age differences in strategy use. It is novel here that the authors are looking at age differences in the felt hand position post learning, but the Vandervoorde paper should be discussed in the introduction to help place the current study better in context. The authors do not well address why their findings might differ from those of previous studies- why did they find differences in explicit strategy use whereas Vandervoorde report implicit model recalibration declines with age?

How were the older adults screened for cognitive status? This is particularly important here given the focus on explicit instructions and cognitive strategies.

How many trials were omitted due to the visual inspection process described on page 15? Please provide some visual examples of these trajectories, as well as for the data of subjects that were excluded.

The authors should be commended for the sample size tested, which is larger than some studies using this paradigm. Was a power analysis conducted, either a priori or post hoc?

The manuscript needs some editing for language and word choice throughout.

6. PLOS authors have the option to publish the peer review history of their article (what does this mean?). If published, this will include your full peer review and any attached files.

Reviewer #1: Yes: Sophia Vinci-Booher

Reviewer #2: No

Reviewer #3: No

---

## [Author Response · Author response to Decision Letter 0]

26 Jul 2020

Please see the attached PDFs for both the revised manuscript and reply to reviewers.

---

## [Decision Letter · Decision Letter 1]

31 Aug 2020

The effect of age on visuomotor learning processes

PONE-D-19-19971R1

Dear Dr. 't Hart,

We’re pleased to inform you that your manuscript has been judged scientifically suitable for publication and will be formally accepted for publication once it meets all outstanding technical requirements.

Kind regards,

Sergiy Yakovenko

Academic Editor

PLOS ONE

Additional Editor Comments (optional):

Thank you for your submission. I am happy to inform you that the reviewers approved this submission for publication.

Reviewers' comments:

Reviewer's Responses to Questions

**Comments to the Author**

1. If the authors have adequately addressed your comments raised in a previous round of review and you feel that this manuscript is now acceptable for publication, you may indicate that here to bypass the “Comments to the Author” section, enter your conflict of interest statement in the “Confidential to Editor” section, and submit your "Accept" recommendation.

Reviewer #1: All comments have been addressed

Reviewer #2: All comments have been addressed

Reviewer #3: All comments have been addressed

2. Is the manuscript technically sound, and do the data support the conclusions?

Reviewer #1: Yes

Reviewer #2: Yes

Reviewer #3: Yes

3. Has the statistical analysis been performed appropriately and rigorously? 

Reviewer #1: Yes

Reviewer #2: Yes

Reviewer #3: Yes

4. Have the authors made all data underlying the findings in their manuscript fully available?

Reviewer #1: Yes

Reviewer #2: Yes

Reviewer #3: Yes

5. Is the manuscript presented in an intelligible fashion and written in standard English?

Reviewer #1: Yes

Reviewer #2: Yes

Reviewer #3: Yes

6. Review Comments to the Author

Reviewer #1: The revisions made to the manuscript clarified the goals of the study and addressed my overall concerns. My minor comments were also addressed.

Reviewer #2: (No Response)

Reviewer #3: The authors have addressed all of my concerns in this revision. The goals of the study are now much more clear.

7. PLOS authors have the option to publish the peer review history of their article (what does this mean?). If published, this will include your full peer review and any attached files.

Reviewer #1: **Yes: **Sophia Vinci-Booher

Reviewer #2: No

Reviewer #3: No

---

## [Editor Report · Acceptance letter]

4 Sep 2020

PONE-D-19-19971R1 

The effect of age on visuomotor learning processes 

Dear Dr. 't Hart:

I'm pleased to inform you that your manuscript has been deemed suitable for publication in PLOS ONE. Congratulations! Your manuscript is now with our production department. 

Kind regards, 

on behalf of

Dr. Sergiy Yakovenko 

Academic Editor

PLOS ONE